# Early Intervention of Pulmonary Rehabilitation for Fibrotic Interstitial Lung Disease Is a Favorable Factor for Short-Term Improvement in Health-Related Quality of Life

**DOI:** 10.3390/jcm10143153

**Published:** 2021-07-16

**Authors:** Satoshi Matsuo, Masaki Okamoto, Tomoyuki Ikeuchi, Yoshiaki Zaizen, Atsushi Inomoto, Remi Haraguchi, Shunichiro Mori, Retsu Sasaki, Takashi Nouno, Tomohiro Tanaka, Tomoaki Hoshino, Toru Tsuda

**Affiliations:** 1Kirigaoka Tsuda Hospital, Kirigaoka 3-9-20, Kokura-kita-ku, Kitakyushu, Fukuoka 802-0052, Japan; matsuos@k-you.or.jp (S.M.); ikeuchit@k-you.or.jp (T.I.); moris@k-you.or.jp (S.M.); sasakir@k-you.or.jp (R.S.); tsudat@k-you.or.jp (T.T.); 2Department of Respirology and Clinical Research Center, National Hospital Organization Kyushu Medical Center, Jigyohama 1-8-1, Chuou-ku, Fukuoka 810-0065, Japan; nouno_takashi@med.kurume-u.ac.jp (T.N.); tanaka_tomohiro@med.kurume-u.ac.jp (T.T.); 3Division of Respirology, Neurology, and Rheumatology, Department of Internal Medicine, Kurume University School of Medicine, Ashahi-machi 67, Kurume, Fukuoka 830-0011, Japan; zaizen_yoshiaki@med.kurume-u.ac.jp (Y.Z.); hoshino@med.kurume-u.ac.jp (T.H.); 4Department of Rehabilitation, Kyushu Nutrition University, Kuzuharatakamatsu 1-5-1, Kokura-minami-ku, Kitakyushu, Fukuoka 800-0298, Japan; inomoto@knwu.ac.jp; 5Department of Rehabilitation, National Hospital Organization Kyushu Medical Center, Jigyohama 1-8-1, Chuou-ku, Fukuoka 810-0065, Japan; haraguchi.remi.sx@mail.hosp.go.jp

**Keywords:** pulmonary rehabilitation, exercise therapy, fibrosing interstitial lung disease, idiopathic pulmonary fibrosis, health-related quality of life

## Abstract

Patients with fibrosing interstitial lung disease (FILD) have poor health-related quality of life (HRQOL). We analyzed predictors of short-term improvement of HRQOL after starting pulmonary rehabilitation (PR) in moderate to severe FILD patients. This study involved 28 consecutive patients with FILD (20 males, median age of 77.5 years), who participated in PR program of our hospital for >6 weeks. The St. George’s Respiratory Questionnaire (SGRQ) score and the 6-min walk distance (6MWD) were evaluated before and after PR, and the predictors of efficacy of PR were analyzed. The duration from diagnosis of FILD to start of PR showed a positive correlation with the increase in the SGRQ score, and the baseline SGRQ score showed a negative correlation with increase in the 6MWD. The FILD subtype, modified Medical Research Council score, and treatment history were not associated with the endpoints. According to the receiver operating characteristic curve (ROC) analyses, starting PR within 514 days after diagnosis of FILD was a significant favorable predictor of improvement in the SGRQ total score more than a minimal clinically important difference of 4. In this study, early intervention of PR and lower SGRQ score were associated with the favorable response to PR. PR for FILD should be initiated early before the disease becomes severe.

## 1. Introduction

Fibrosing interstitial lung diseases (FILDs) are a group of diseases with heterogeneous etiologies such as connective tissue disease and environmental exposure. The major subtype of FILD is idiopathic interstitial pneumonia (IIP), which is characterized by an unknown etiology [1]. Patients with FILD show heterogeneous clinical behaviors, and the patients who develop progressive lung fibrosis are termed with progressive fibrosing interstitial lung disease (PF-ILD) [2,3]. Patients with PF-ILD have reduced health-related quality of life (HRQOL) and survival expectancy [1,2,3]. Idiopathic pulmonary fibrosis (IPF), a pathologically usual interstitial pneumonia, is the most common form of IIP [4,5,6]. The estimated prevalence of IPF reportedly ranges from 2 to 43 cases per 100,000 people globally and 10 cases per 100,000 people in Japan [4,7]. Patients with IPF have a poorer prognosis caused by disease progression than other FILD [1,4,5,6]. Patients with FILD often experience impairment in their daily activities because of dyspnea on exertion, depression, and malnutrition with disease progression [8,9,10,11,12]. HRQOL questionnaires such as the Saint George’s Respiratory Questionnaire (SGRQ), the degree of dyspnea, and exercise tolerance are associated with one another and survival in patients with FILD [13,14,15,16,17]. Some randomized controlled trials (RCTs) and prospective or retrospective studies have suggested that pulmonary rehabilitation (PR) is able to contribute to improvement of symptoms, exercise tolerance, and HRQOL in patients with FILD [18,19,20,21,22,23,24,25,26,27,28,29]. PR may also contribute to reducing mortality in patients with FILD, because some studies showed that the SGRQ score, modified Medical Research Council (mMRC) score, lowest SpO_2_, and 6-min walk distance (6MWD) were associated with survival in patients with IPF [15,16,17]. The consensus on the short-term efficacy of PR for patients with FILD has been described in previous guidelines and reviews [5,6,8,9,10,18]. However, many studies showing the efficacy of PR have targeted patients with mild to moderate FILD and have not revealed the efficacy of PR in those with severe FILD [20,21,22,23,24,25,26,27,28,29]. A milder degree of dyspnea and higher pulmonary function, exercise tolerance, and HRQOL at baseline were associated with higher efficacy of PR in some previous studies of PR [21,23,24,26,29]. However, no studies have analyzed the most appropriate intervention timing of PR for patients with FILD. In this study, we analyzed the predictors of the short-term efficacy of PR in patients with FILD, including the PR intervention timing. In addition, previous studies of PR had analyzed patients with mild to moderate FILD [20,21,22,23,24,25,26,27,28,29]. Our study focuses on the efficacy of PR for patients with more severe FILD in the previous studies.

## 2. Patients and Methods

### 2.1. Patients

We analyzed 28 consecutive patients with FILD (20 males, median age of 77.5 years) of unknown etiology (so-called IIP) who participated in the PR program of our hospital, a specialized institution for PR, for >6 weeks as in-hospital or ambulatory treatment from October 2013 to October 2020. All patients with FILD were diagnosed and classified by two well-experienced respiratory physicians specializing in diffuse lung disease based on the global guidelines for IIP [1,4,6]. Patients who were diagnosed with secondary FILD such as hypersensitivity pneumonia, connective tissue disease, or pneumoconiosis were excluded from this study. Similarly, patients who began PR during merging acute exacerbation of FILD, infectious disease, and heart failure were excluded. Patients with malignant disease uncontrolled by curative treatment were also excluded. All patients in this study underwent examinations at the start and end of PR therapy, including the 6-min walk test (6MWT), evaluation of symptoms, and assessment of HRQOL as measured by the mMRC and SGRQ total scores in accordance with previously described methods [13,30,31].

### 2.2. Pulmonary Rehabilitation Program

All patients participated in a standard PR program at our hospital. PR for FILD patients was usually started when dyspnea in patients worsened. We considered worsening of dyspnea if the mMRC grade increases by 1 or more or if the patient becomes aware of worsening after the second visit compared to the first visit. The outpatient program was provided twice a week for 60 min each. The content of PR was determined according to the baseline evaluation and consisted of instruction and education in breathing techniques and supervised exercise (strength training, treadmill, and bicycle ergometer) in accordance with the standard protocol previously reported [8,23]. The inpatient program was provided 5 days a week with content similar to that in the outpatient program. PR was performed for 6 to 10 weeks in both outpatient and inpatient. The oxygen dose for patients receiving long-term oxygen therapy (LTOT) during exercise therapy was similar to the usual treatment dose.

### 2.3. Pulmonary Function Tests

Pulmonary function tests, including forced vital capacity (FVC) and forced expiratory volume in 1 s, were performed using an electrical spirometer according to the guidelines of the American Thoracic Society/European Respiratory Society [32].

### 2.4. Statistical Analyses

Data were expressed as median (25th–75th percentiles of interquartile range). The present study analyzed a small population that is unlikely to be normally distributed, so we performed a nonparametric statistical analysis. We analyzed comparisons and correlations between two clinical parameters using the Mann–Whitney U test and Spearman’s rank correlation coefficient. The clinical parameters before and after PR were compared using the Wilcoxon signed-rank test. The mMRC score cut-off value (grade 3) and the minimal clinically important difference (MCID) of the 6MWD (24 m) and SGRQ total score (4 points) were set in accordance with the previous reports [16,17,24,33,34]. The cut-off levels of continuous parameters were determined as those with the highest Youden index (i.e., sensitivity + specificity − 1) by receiver operating characteristic (ROC) curve analysis [35]. A *p*-value < 0.05 was considered statistically significant. All statistical analyses were performed using SPSS version 27.0 (IBM Corp., Armonk, NY, USA).

## 3. Results

### 3.1. Patient Characteristics

The patient characteristics are shown in Table 1. Among 28 patients with FILD, 21 (75%) were diagnosed with IPF and the remaining 7 (25%) were diagnosed with unclassifiable IIP. The median duration from diagnosis of FILD to the start of PR was 656.5 (480.0–1240.8) days. PR was performed in all patients for >6 weeks, and the median duration of PR was 55.5 (42.0–74.0) days. The median FVC, 6MWD, and SGRQ total score were 58.2 (48.3–73.9) percent predicted, 266.0 (236.2–315.8) m, and 53.0 (37.7–68.9), respectively. Among 28 patients with FILD, 9 (32%) received treatment with oral corticosteroids, 7 (25%) received treatment with anti-fibrotic agents, and 9 (32%) received LTOT.

### 3.2. Correlations among Baseline Clinical Data

The correlations among the baseline clinical data are shown in Table 2. Both the FVC (Spearman’s r = −0.59, *p* = 0.0030) and 6MWD (Spearman’s r = −0.48, *p* = 0.018) were significantly correlated with the SGRQ total score. Age, duration from diagnosis of FILD to start of PR, and body mass index (BMI) showed no significant correlation with either the 6MWD or SGRQ total score at baseline. The results of comparison between two groups by the Mann–Whitney U test are shown in Appendix A. Among the 28 patients with FILD, the 6MWD was lower in patients with non-IPF than in those with IPF (225.0 and 280.0 m, respectively; *p* = 0.012), and in patients with mMRC grades 3 and 4 than in those with mMRC grades 0, 1, and 2 (244.0 and 311.5 m, respectively; *p* = 0.022). The SGRQ total score was higher in patients with mMRC grades 3 and 4 than in those with grades 0, 1, and 2 (65.0 and 38.0, respectively; *p* = 0.0050) and in patients who did than did not receive LTOT (70.9 and 46.0, respectively; *p* = 0.020). Treatment with neither oral corticosteroids nor anti-fibrotic agents was associated with the SGRQ total score or 6MWD at baseline.

### 3.3. Analyses of Predictors of Efficacy of PR

The results of the comparison between the 6MWD and SGRQ total score for analyzing the short-term efficacy of PR are shown in Table 3. There was no improvement in the 6MWD (266.0 and 271.5 m, *p* = 0.26) or SGRQ total score (53.0 and 51.4, *p* = 0.62) between before and after PR among all 28 patients with FILD.

We analyzed whether baseline clinical data are correlated with change of 6MWD and SGRQ total score after PR for detecting the predictor of the short-term efficacy of PR (Table 4). The duration from diagnosis of FILD to the start of PR showed a significant positive correlation with the increase in the SGRQ total score after PR (Spearman’s r = 0.58, *p* = 0.0058). The SGRQ total score at baseline showed a significant negative correlation with the increase in the 6MWD after PR (Spearman’s r = −0.46, *p* = 0.025). The BMI tended to show a positive correlation with the increase in the 6MWD after PR (Spearman’s r = 0.34, *p* = 0.080). Age, FVC, and 6MWD at baseline and duration of PR showed no significant correlation with either the change in the 6MWD or SGRQ total score after PR. We compared the change in the 6MWD and SGRQ total score after PR between the two groups classified by their clinical data at baseline (Appendix A). Treatment with anti-fibrotic agents tended to be associated with a greater increase in the 6MWD (40.0 vs. 5.0 m, *p* = 0.087). There was no difference of increase in the 6MWD and the SGRQ total score in between male and female (8.5 vs. 8.5 m, *p* = 0.63 and −2.6 vs. 3.8 points, *p* = 0.29). Moreover, ILD subtype, mMRC score, and therapy at baseline were not associated with either the change in the 6MWD or SGRQ total score after PR. According to the above results, a shorter duration from diagnosis of FILD to the start of PR and a lower baseline SGRQ total score can predict better response to PR. 

The results of the ROC analyses are shown in Table 5. The ROC curves of clinical data that were associated with the improvement over the MCID of the SGRQ total score or 6MWD are shown in Figure 1. Less than cut-off level of the duration from diagnosis of FILD to the start of PR was significantly associated or tended to be associated with improvement over the MCID of the SGRQ total score after PR (MCID = 4 points, sensitivity = 0.93, specificity = 0.71, area under ROC curve = 0.86, *p* = 0.0074), and 6MWD (MCID = 24 m, sensitivity = 0.72, specificity = 0.60, area under ROC curve = 0.64, *p* = 0.087).

## 4. Discussion

In the present study, we analyzed the clinical factors that may predict short-term efficacy of PR for patients with more severe FILD than those in previous reports in a specialized institution for PR. Among the patients in the 10 reports of PR [20,21,22,23,24,25,26,27,28,29], the median FVC at baseline was <60% in only one report and ranged from 54.1% to 78% predicted, and the median 6MWD at baseline was 308 to 526 m; these values were higher than those in the present study. We found that a shorter duration from diagnosis of FILD to start of PR was associated with improvement in the SGRQ total score, and the lower SGRQ total score at baseline was associated with improvement in 6MWD. In the present study, gender was not associated with the change in 6MWD and SGRQ total score. Previous studies of PR for patients with FILD suggested that a higher FVC, a milder exercise-induced hypoxemia, and a lower mMRC and SGRQ total score, and right ventricular systolic pressure were associated with better response to PR [21,23,24,26]. However, there is no established predictor that has shown reproducible data. Nishiyama et al. [21] performed a RCT of 10-week PR for 28 patients with IPF who had a median FVC of 66.1% predicted. The authors found that a lower SGRQ at baseline was associated with improvement of the 6MWD, similar to the result of the present study. PR for FILD should be started at the early phase before the disease becomes severe in accordance with the finding that lower severity of FILD is associated with a better response to PR [21,23,24,26,29]. In addition, Wallaert et al. [27] performed a prospective study of 2-month PR for 112 patients with FILD and found that patients who dropped out had lower pulmonary function at baseline than those who completed the study. Most of those previous reports did not show the duration from diagnosis of FILD to start of PR, and no study has analyzed the association between this duration and the efficacy of PR. The results of the present study suggest that the intervention of PR for FILD should not be delayed from diagnosis in order to maintain the therapeutic effect. Early intervention of PR before decrease of exercise tolerance due to disease progression is also important in respiratory diseases other than IPF. On the other hand, in IPF patients, exercise-induced hypoxemia is often more severe than other respiratory diseases, so early intervention for PR is more important [8,9]. Moreover, improvement of the method of PR itself is necessary for patients with severe FILD. PR for those patients should be started with conditioning including stretching and relaxation or low-intensity endurance and strength training [8,9].

Among all 28 subjects analyzed in the present study, no significant improvement in the 6MWD or SGRQ total score was observed after PR. Although the efficacy of PR in patients with FILD is reportedly lower than that in patients with chronic obstructive pulmonary disease [8,9,25], previous RCTs and prospective and retrospective studies have shown significant improvements in the 6MWD, HRQOL score, and dyspnea score after PR for FILD [20,21,22,23,24,25,26,27,28,29]. The reason for the difference in the efficacy of PR for FILD between the present study and previous studies is unknown. It is possible that patients with more severe conditions were selected in the present study than in previous studies of PR, which may have influenced the outcome after PR.

Among the 28 patients with FILD in this study, there was no difference in the efficacy of PR between patients with IPF and non-IPF. This result is consistent with previous reports [20,23]. Some previous reports have shown the short-term efficacy of PR for FILD but denied the long-term effects at ≥6 months [8,9,20,23,25,26]. However, Vainshelboim et al. [22] showed improvement in exercise tolerance at 11 months after 12-week PR in a RCT involving 34 patients with IPF. Wallaert et al. [27] also showed improvement in exercise tolerance and HRQOL 12 months after the start of PR for FILD in a prospective study. These results were shown after establishment of treatment with anti-fibrotic agents including pirfenidone or nintedanib for patients with IPF in phase 3 trials and coverage of these agents by insurance [36,37]. A recent phase 3 trial showed the efficacy of nintedanib for non-IPF PF-ILD [38]. The therapeutic effects of anti-fibrotic agents for FILD involve suppression of the decrease in FVC, HRQOL, deterioration of dyspnea, or development of acute exacerbations as well as improved survival, but these agents are not curative [36,37,38,39]. Because treatment with anti-fibrotic agents has the potential to improve the survival of patients with FILD [39], starting PR at an appropriate time is important for long-term maintenance of the patient’s HRQOL during the clinical course. In the prospective study by Wallaert et al., the home-based pulmonary rehabilitation program has been performed for 12 months [27]. In this program, each weekly session was conducted under the direct supervision of a team member, but patients were expected to perform a personalized endurance physical exercise plan, unsupervised, on the other days of the week. Various methods should be considered such as remote control of PR by patients’ self-management for maintaining the effect of PR on FILD for a long period of time. Moreover, a prospective study should be performed to evaluate the long-term efficacy of PR for FILD patients.

The present study had some limitations. First, the sample size was small, mainly because the study was conducted at only one center. Second, the study design was retrospective. Further prospective studies with larger populations will be needed for clarifying some issues including more accurate time of intervention of PR and effect of treatment with antifibrotic agent on the efficacy of PR. Third, the optimal intervention timing may have diversity depending on the heterogeneous clinical behavior in FILD patients. Subgroup-specific analysis with different clinical behavior will be required. Fourth, the present study did not clarify the predictors of long-term efficacy of PR. We will analyze this issue in a further study.

## 5. Conclusions

In this study, early intervention of PR after diagnosis of FILD was associated with improved HRQOL, and lower SGRQ total score were associated with an improved 6MWD. PR for FILD should be initiated early before the disease becomes severe.

## Figures and Tables

**Figure 1 jcm-10-03153-f001:**
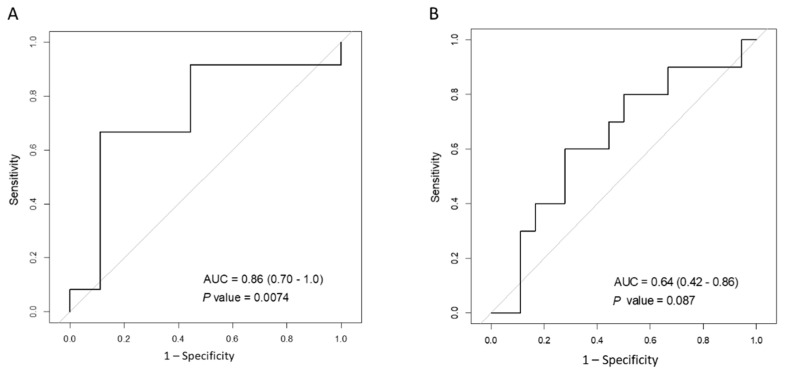
ROC curve for predicting improvement of more than minimal clinically important difference of (**A**) SGRQ total score (4 points) and (**B**) 6MWD (24 m) by duration from diagnosis of FILD to start of PR. ROC, receiver operating characteristic; AUC, area under curve; SGRQ, St. George’s Respiratory Questionnaire; 6MWD, 6-min walk distance; FILD, fibrosing interstitial lung disease; PR, pulmonary rehabilitation.

**Table 1 jcm-10-03153-t001:** All patient baseline characteristics.

Patient Characteristics	*n* = 28
Age, yr	77.5 (75.0–81.0)
Gender: Male/Female	20 (71)/8 (29)
FILD subtype, IPF/Non-IPF	21 (75)/7 (25)
Duration from diagnosis to PR, days	656.5 (480.0–1240.8)
Duration of PR, days	55.5 (42.0–74.0)
BMI, kg/m^2^	20.1 (18.2–24.5)
FVC, %predicted	58.2 (48.3–73.9)
mMRC grade, 0/1/2/3/4	0 (0)/7 (25)/5 (18)/9 (32)/7 (25)
6MWD, m	266.0 (236.2–315.8)
SGRQ total score	53.0 (37.7–68.9)
Therapy at baseline	
Corticosteroid	9 (32)
Anti-fibrotic agent	7 (25)
LTOT	9 (32)

Data are presented as median (interquartile range) for continuous variables and number (%) for categorical variables. FILD, fibrotic interstitial lung disease; IPF, idiopathic pulmonary fibrosis; PR, pulmonary rehabilitation; BMI, body mass index; FVC, forced vital capacity; mMRC, modified Medical Research Council; 6MWD, 6-min walk distance; SGRQ, St. George’s Respiratory Questionnaire; LTOT, long-term oxygen therapy.

**Table 2 jcm-10-03153-t002:** Relationship of baseline values with 6MWD and SGRQ total score.

	6MWD, m	*p* Value	SGRQ Total	*p* Value
Age, yr	−0.25	0.20	−0.26	0.21
Duration from diagnosis to PR, days	−0.08	0.67	0.27	0.21
BMI, kg/m^2^	0.08	0.70	−0.34	0.10
FVC, %predicted	0.25	0.20	−0.59	0.0030 *
6MWD, m	-	-	−0.48	0.018 *
SGRQ total score	−0.48	0.018 *	-	-

Data are presented as Spearman’s rho. PR, pulmonary rehabilitation; BMI, body mass index; FVC, forced vital capacity; 6MWD, 6-min walk distance; SGRQ, St. George’s Respiratory Questionnaire. * *p* < 0.05.

**Table 3 jcm-10-03153-t003:** Changes in 6MWD and SGRQ total score.

	Pre PR	Post PR	*p* Value
6MWD, m	266.0 (236.2–315.8)	271.5 (219.2–320.0)	0.26
SGRQ total score	53.0 (37.7–68.9)	51.4 (30.7–67.1)	0.62

Data are presented as median (interquartile range). PR, pulmonary rehabilitation; 6MWD, 6-min walk distance; SGRQ, St. George’s Respiratory Questionnaire.

**Table 4 jcm-10-03153-t004:** Relationship between baseline values and response to PR.

	Δ6MWD		ΔSGRQ Total	
	Spearman’s R	*p* Value	Spearman’s R	*p* Value
Age, yr	−0.15	0.44	0.022	0.92
Duration from diagnosis to PR, days	−0.31	0.11	0.58	0.0058 *
Duration of PR, days	0.15	0.43	0.36	0.11
BMI, kg/m^2^	0.34	0.080	−0.36	0.11
FVC, %predicted	0.12	0.53	0.050	0.82
6MWD, m	0.21	0.29	0.052	0.82
SGRQ total score	−0.46	0.025 *	0.16	0.50

Data are presented as Spearman’s rho. PR, pulmonary rehabilitation; BMI, body mass index; FVC, forced vital capacity; 6MWD, 6-min walk distance; SGRQ, St. George’s Respiratory Questionnaire. * *p* < 0.05.

**Table 5 jcm-10-03153-t005:** Area under the curve for predicting improvement over MCID of (**A**) SGRQ total score and (**B**) 6MWD.

A. SGRQ Total Score.					
	Cut-Off Level	AUC (95% CI)	Sensitivity	Specificity	*p* Value
Age, yr	77.5	0.53 (0.21–0.85)	0.71	0.71	0.82
Duration from diagnosis to PR, days	514.5	0.86 (0.70–1.0)	0.93	0.71	0.0074 *
Duration of PR, days	54.0	0.70 (0.42–0.98)	0.71	0.86	0.14
BMI, kg/m^2^	24.8	0.55 (0.26–0.84)	0.86	0.43	0.71
FVC, %predicted	40.9	0.57 (0.29–0.86)	0.93	0.29	0.60
6MWD, m	324.0	0.51 (0.18–0.84)	0.86	0.43	0.94
SGRQ total score	37.3	0.59 (0.25–0.93)	0.93	0.57	0.50
**B. 6MWD.**					
Age, yr	75.5	0.46 (0.24–0.68)	0.33	0.80	0.83
Duration from diagnosis to PR, days	582.0	0.64 (0.42–0.86)	0.72	0.60	0.087
Duration of PR, days	52.5	0.68 (0.47–0.90)	0.56	0.80	0.12
BMI, kg/m^2^	24.5	0.64 (0.41–0.87)	0.89	0.50	0.23
FVC, %predicted	45.2	0.59 (0.35–0.83)	0.89	0.40	1.0
6MWD, m	336.5	0.64 (0.40–0.64)	1.0	0.40	0.31
SGRQ total score	35.2	0.65 (0.42–0.89)	0.93	0.40	0.18

MCID, minimal clinically important difference; AUC, area under receiver operating characteristic curve; CI, confidence interval; PR, pulmonary rehabilitation; BMI, body mass index; FVC, forced vital capacity; 6MWD, 6-min walk distance; SGRQ, St. George’s Respiratory Questionnaire. * *p* < 0.05.

## Data Availability

The data presented in this study are available on request from the corresponding author. The data are not publicly available due to the ethical considerations.

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
