# Peer review of "Early Intervention of Pulmonary Rehabilitation for Fibrotic Interstitial Lung Disease Is a Favorable Factor for Short-Term Improvement in Health-Related Quality of Life"

_jcm, 2021, doi:10.3390/jcm10143153_

Round 1
Reviewer 1 Report
Dear authors,
I review your review regarding early intervention of pulmonary rehabilitation for fibrotic interstitial lung diseases and I would like the authors to clarify some aspects:
- add in the methods for how long the PR program takes care and when do you usually start doing it after the diagnosis.
- include in the results and discussion section if the sex seems to have an effect in your results
- discuss if a daily routine and not only two times per week it will possible and if the authors think that this might improve also the outcome
- is possible that the patients perform this PR on their own? If yes, it might be possible to train them for 6 weeks and then do a longer follow up on them to check again the outcome? I think any kind of future possible intervetion, usge of the PR it should be mention as future perspective in the discussion.
Author Response
Response to Comments
Reviewer 1
I review your review regarding early intervention of pulmonary rehabilitation for fibrotic interstitial lung diseases and I would like the authors to clarify some aspects:
add in the methods for how long the PR program takes care and when do you usually start doing it after the diagnosis.
include in the results and discussion section if the sex seems to have an effect in your results
discuss if a daily routine and not only two times per week it will possible and if the authors think that this might improve also the outcome
is possible that the patients perform this PR on their own? If yes, it might be possible to train them for 6 weeks and then do a longer follow up on them to check again the outcome? I think any kind of future possible intervetion, usge of the PR it should be mention as future perspective in the discussion.
Comment
Thank you for some important comments.
We added the duration of PR program and the time of starting PR in the Methods.
We consider that gender did not influence the endpoint in the present study, because gender was not associated with the change in 6MWD and SGRQ. We have added the following sentences in the Results and Discussion.
As the reviewer suggested, it should be considered various methods for maintaining the effect of PR on FILD for a long period of time. We have added discussion of those issues as follows.
Line 86 (Methods)
PR for FILD patients was usually started when dyspnea in patients worsened.
Line 92 (Methods)
PR was performed for 6 to 10 weeks in both outpatients and inpatients.
Line 169 (Results)
There is no difference of increase in the 6MWD and the SGRQ in between male and female (8.5 vs 8.5 m, P = 0.63 and -2.6 vs 3.8 points, P = 0.29).
Line 206 (Discussion)
In the present study, gender was not associated with the change in 6MWD and SGRQ.
Line 257 (Discussion)
In the prospective study by Wallaert et al., the home-based pulmonary rehabilitation program has been performed for 12 months [27]. In this program, each weekly session was conducted under the direct supervision of a team member, but patients were expected to perform a personalized endurance physical exercise plan, unsupervised, on the other days of the week. It should be considered various methods such as remote control of PR by patient's self-management for maintaining the effect of PR on FILD for a long period of time. Moreover, a prospective study should be performed to evaluate the long-term effects of PR for FILD patients.

Reviewer 2 Report
General comments
The authors have explored timing and treatment synergies for pulmonary rehabilitation to combat primary fibrotic interstitital lung disease, a rare disease with bad prognosis. The challenge is well-established within the body of knowledge described in the Introduction. The study has been carried out on a carefully selected patient cohort that mostly exclude confounding variables. The patients’ status was tracked through appropriate quality parameters. The utilization of statistics is correct although the size of the dataset was (maybe unavoidably) rather small. I find results are quite useful and of high interest for the medical community that treat this and related diseases all over the globe.
1) As recognized by authors, the patient cohort size is limiting. It is not always possible, but authors should consider increase the patient cohort in future similar studies to reach statistical significance on those issues that are suspicious to be correlated (e.g. in lines 165-6 and 180-1). I encourage them to start inter-center studies, which may reveal such hidden data associations in low-prevalence diseases like this just by a question of data amount.
2) How does PR perform in severe cases of other lung-related diseases? Is FILD the norm or the exception on this matter? Please, discuss.
3) I wonder whether there is room for improvement of the PR itself, in particular regarding patients with severe FILD. Please, discuss.
Author Response
Response to Comments
Reviewer 2
Comments
1) As recognized by authors, the patient cohort size is limiting. It is not always possible, but authors should consider increase the patient cohort in future similar studies to reach statistical significance on those issues that are suspicious to be correlated (e.g. in lines 165-6 and 180-1). I encourage them to start inter-center studies, which may reveal such hidden data associations in low-prevalence diseases like this just by a question of data amount.
Response
Thank you for your very important comment. As the reviewer suggested, the results in the present study need to be reanalyzed in studies with a larger population. Their analyses may reveal more accurate time of PR initiation and the effects of treatment with antifibrotic agent on the effects of PR. We have added the following sentence to the limitation part.
Line 268 (Discussion)
Further prospective studies with larger populations will be needed for clarifying some issued including more accurate time of intervention of PR and effect of treatment with antifibrotic agent on the efficacy of PR.
Comment
2) How does PR perform in severe cases of other lung-related diseases? Is FILD the norm or the exception on this matter? Please, discuss.
Response
Thank you for your important comment. Even in respiratory diseases other than IPF, the decrease in exercise tolerance due to disease progression may reduce the efficacy of PR, and early intervention of PR may be necessary. On the other hand, in IPF patients, exercise-induced hypoxemia is often more severe than other diseases, so early intervention of PR is more important. We have added the following description for suggesting the above discussion.
Line 223(Discussion)
Early intervention of PR before decrease of exercise tolerance due to disease progression is also important in respiratory diseases other than IPF. On the other hand, in IPF patients, exercise-induced hypoxemia is often more severe than other respiratory diseases, so early intervention for PR is more important [8, 9].
Comment
3) I wonder whether there is room for improvement of the RP itself, in particular regarding patients with severe FILD. Please, discuss.
Response
Thank you for your important comment. As the reviewer suggested, improvement of the PR method for patients with severe FILD. We have added the following sentence in the text.
Line 226 (Discussion)
Moreover, improvement of the method of PR itself is necessary for patients with severe FILD. PR for those patients should be started with conditioning including stretching and relaxation or low-intensity endurance and strength training [8,9].

Reviewer 3 Report
In the underlying manuscript the authors report the results of a small retrospective study in which they looked at the effect of pulmonary rehabilitation in patients with severe fibrotic ILD. The authors show that earlier start of PR upon diagnosis is beneficial. There are some weaknesses in the study, but they are mostly addressed in the discussion of the manuscript. I would like to encourage the authors to perform analysis on the long-term effects of PR related to start upon diagnosis as soon as possible.
I only have some minor comments.
In the introduction the authors state that the median survival of IPF patients is 2 to 5 years. This is probably data from a few years ago as life expectancy has increased in this population since the use of nintedanib and pirfinedone has become standard treatment of care.
There are some minor grammatical/spelling errors throughout the manuscript. For instance:
- line 203: ‘nadir’ For me it is unclear what the authors want to say here.
Author Response
Response to Comments
Reviewer 3
Comment
In the introduction the authors state that the median survival of IPF patients is 2 to 5 years. This is probably data from a few years ago as life expectancy has increased in this population since the use of nintedanib and pirfinedone has become standard treatment of care.
Response
Thank you for your comment. As the reviewer suggested, overall survival of IPF patients is likely to be improved than before, because some registry studies and propensity score matching analyzes have shown that IPF patients who were treated with antifibrotic agents have longer survival than those who were not. We should not show the survival of IPF before anti-fibrotic agents are covered by insurance in this treatise in this paper. We revised the sentence including survival of IPF as follows.
Line 48 (Introduction)
Patients with IPF have a poorer prognosis than other FILD [1, 4–6].
Comment
There are some minor grammatical/spelling errors throughout the manuscript. For instance:
- line 203: ‘nadir’ For me it is unclear what the authors want to say here.
Response
Thank you for your comment. We have changed the sentences including SpO2 nadir as follows.
As the reference was not shown in this sentence, I have added it.
Line 208 (Discussion)
Previous studies of PR for patients with FILD suggested that a higher FVC, a milder exercise-induced hypoxemia and a lower mMRC and SGRQ score, and right ventricular systolic pressure were associated with better response to PR [21, 23-24, 26].
